# Description and Characterization of a Novel Human Mast Cell Line for Scientific Study

**DOI:** 10.3390/ijms20225520

**Published:** 2019-11-06

**Authors:** Arnold S. Kirshenbaum, Yuzhi Yin, J. Bruce Sundstrom, Geethani Bandara, Dean D. Metcalfe

**Affiliations:** 1Mast Cell Biology Section, Laboratory of Allergic Diseases, Bethesda, MD 20892, USA; yiny@niaid.nih.gov (Y.Y.); bandarag@niaid.nih.gov (G.B.); dmetcalfe@niaid.nih.gov (D.D.M.); 2AIDS Research Review Branch/SRP, National Institute of Allergy and Infectious Diseases, National Institutes of Health, Bethesda, MD 20892, USA; sundstromj@niaid.nih.gov

**Keywords:** mast cell, LAD cells, HIV, allergy, FcεR1, stem cell factor, c-*KIT*, SOCS, *MYC*

## Abstract

Background: Laboratory of allergic diseases 2 (LAD2) human mast cells were developed over 15 years ago and have been distributed worldwide for studying mast cell proliferation, receptor expression, mediator release/inhibition, and signaling. LAD2 cells were derived from CD34+ cells following marrow aspiration of a patient with aggressive mastocytosis with no identified mutations in *KIT*. Another aspiration gave rise to a second cell line which has recently been re-established (LADR). We queried whether LADR had unique properties for the preclinical study of human mast cell biology. Methods: LADR and LAD2 cells were cultured under identical conditions. Experiments examined proliferation, beta-hexosaminidase (β-hex) release, surface receptor and granular protease expression, infectivity with HIV, and gene expression. Results: LADR cells were larger and more granulated as seen with Wright–Giemsa staining and flow cytometry, with cell numbers doubling in 4 weeks, in contrast to LAD2 cells, which doubled every 2 weeks. Both LADR and LAD2 cells released granular contents following aggregation of FcεRI. LADR cells showed log-fold increases in FcεRI/CD117 and expressed CD13, CD33, CD34, CD63, CD117, CD123, CD133, CD184, CD193, and CD195, while LAD2 cells expressed CD33, CD34, CD63, CD117, CD133, CD193 but not CD13, CD123, CD184, or CD195. LADR tryptase expression was one-log-fold increased. LADR cell and LAD2 cell chymase expression were similar. Both cell lines could be infected with T-tropic, M-tropic, and dual tropic HIV. Following monomeric human IgE stimulation, LADR cells showed greater surface receptor and mRNA expression for CD184 and CD195. Expression arrays revealed differences in gene upregulation, especially for the suppressor of cytokine signaling (SOCS) family of genes with their role in JAK2/STAT3 signaling and cellular myelo**c**ytomatosis oncogene (c-MYC) in cell growth and regulation. Conclusions: LADR cells are thus unique in that they exhibit a slower proliferation rate, are more advanced in development, have increased FcεRI/CD117 and tryptase expression, have a different profile of gene expression, and show earlier infectivity with HIV-BAL, LAV, and TYBE when compared to LAD2 cells. This new cell line is thus a valuable addition to the few FcεRI+ human mast cell lines previously described and available for scientific inquiry.

## 1. Introduction

Mast cells are known to play a major role in innate and acquired immunity. Since 1988, a number of cell lines have been described, including HMC-1 (Human Mast Cell leukemia-1), LAD2 (Laboratory of Allergic Diseases 2), LUVA (Laboratory of University of VirginiA), and ROSA cell lines [1,2,3,4]. HMC-1 and LAD2 cell lines were derived from patients with mast cell leukemia. HMC 1.1 and 1.2 express *KIT* mutations, while LAD2 cells do not express mutations in *KIT*. By contrast, LUVA and ROSA^KIT WT^ cell lines were derived from CD34+ cells of non-mastocytosis donors and do not express *KIT* mutations. ROSA^KIT D816V^ cells do exhibit activating *KIT* mutations. All cell lines are available for study of preclinical human mast cell biology in lieu of primary mast cell cultures derived from bone marrow, or peripheral or cord blood precursors [5]. The LAD2 mast cell line closely resembles primary human mast cell cultures which are slow growing, have growth dependence on the presence of stem cell factor (SCF), bear functional surface FcεRI receptors, and have the ability to degranulate to immunologic stimuli. LAD2 human mast cells have been maintained in our laboratory for use and distribution worldwide.

LAD2 cells were derived from CD34+ cells following marrow aspiration of a patient with aggressive mastocytosis where mutations in *KIT* were not identified. A second aspiration preserved under liquid N_2_ gave rise to a second mast cell line LAD1, now re-established and characterized and which we term LADR. As will be shown, LADR cells share some similarities to LAD2 cells while differing in some important aspects of degranulation, surface receptor expression, protease content, gene expression, and susceptibility to infection.

## 2. Results

We first expanded and then characterized LADR cells after removing them from liquid nitrogen and testing for cell viability. In culture, LADR cells were larger, more granulated, and slower to proliferate (Figure 1A,B), suggesting a more advanced and mature cell line. LADR granular content of tryptase was a log-fold higher compared to LAD2 cells (Figure 1C). LADR cells stained for granular chymase as has been reported for LAD2 cells (Figure 1D). Degranulation and β-hex release surpassed that of LAD2 cells (Figure 1E). Flow cytometry studies confirmed the larger size (FSC) and increased granularity (SSC) (Figure 2A) of LADR cells. All LADR cells stained positive for CD117 and FcεRI, with increased expression of CD117 and FcεRI when compared to LAD2 cells (Figure 2B,C). As shown in Table 1, cell surface markers showed the added presence on LADR cells of CD13, CD123, and complement receptors CD184 and CD195 are consistent with HIV studies, as will be shown.

Studies with LAD2 cells previously revealed that binding of monomeric human IgE to FcεRI could affect HIV infectivity following upregulation of T- and M-tropic complement co-receptors [6]. Thus, to check for LADR/LAD2 functional differences, LADR and LAD2 cells were initially incubated with monomeric human IgE and complement receptors were studied. As predicted, earlier and higher surface expression of CD184 was noted for LADR cells at 48 h (Figure 3A), consistent with higher mRNA expression at 48 h (Figure 3B, compared with 3A). When infected with HIV with different tropisms, HIV-BAL, HIV-LAV, and HIV-TYBE infectivity, as measured by p24 assay, were detected in LADR cells earlier (day 8) than LAD2 cells (day 12), consistent with their differential surface expression of CD184 (Figure 3C).

Differential gene expression with a heat map display (Figure 4A) showed distinctly upregulated genes when comparing LADR with LAD2 cells. As many as 400 genes were shown to be upregulated (Figure 4B). When compared with LAD2 cells, major canonical pathway differences, reflecting gene upregulation, were noted for LADR cells (Figure 5A). Examples of upregulated genes included SOCS (suppressor of cytokine signaling) with its role in JAK2/STAT3 signaling and cellular myelo**c**ytomatosis oncogene (c-MYC) with its central role in regulation of cell growth, survival, and differentiation (Figure 5B).

## 3. Discussion

In 2003, our laboratory published a report of the LAD2 human mast cell line which offered a unique opportunity to examine the biology of human mast cells and our group has made this cell line available to researchers everywhere. This area of research has gradually matured, and the current interest in disease phenotypes with gain-of-function genetic lesions led to the question whether characterization of additional mast cell lines may show differences in gene expression and provide insight into upregulation and control of mast cell activation, proliferation, and chemotaxis. We thus expanded and characterized LADR cells and compared this line to existing LAD2 cells in our laboratory. LADR cells were larger and more granulated as evidenced by histochemistry and flow cytometry and were slower to proliferate in culture (Figure 1 and Figure 2), suggesting a more advanced and mature cell line. LADR cells, similar to their LAD2 counterparts, survived for weeks in the absence of SCF, but did not proliferate. Degranulation and β-hex release surpassed that of LAD2 cells initially in culture. The LADR granular content of tryptase was a log-fold higher (Figure 1), chymase was similarly expressed, and flow cytometry studies (Figure 2) showed consistent but increased dual staining for CD117 and FcεRI. We previously reported that incubation with monomeric IgE may affect LAD cell expression of CD4 and co-receptors [6]. Functional differences were predicted when earlier and higher expression of CD184 and CD193 was noted for LADR cells following monomeric human IgE incubation (Figure 3). Thus, when LADR cells were infected with HIV with different tropisms, HIV-BAL, HIV-LAV, and HIV-TYBE infectivity, as measured by p24 assay, were detected earlier than LAD2 cells, consistent with their differential surface expression of complement receptors. Previous flow cytometry experiments using CD184 antibody may not have been sensitive enough to detect low level expression of CD184 on LADR at 24 h in the presence of IgE, in contrast to detection of CD184 expression using newer conjugates and LSRFortessa flow cytometry. Gene analysis comparing LADR with LAD2 cells showed a large differential expression of upregulated genes (Figure 4) and canonical pathway differences (Figure 5), which are now being investigated for physiologic importance and impact on mast cell functions.

Our purpose in studying additional LAD cell lines was to compare the characteristics of LADR with those published for LAD2 and determine advantages for using one cell line over the other. To this end, LADR cells offer clear advantages depending on both the research at hand and the human mast cell biology questions being addressed. Compared with LAD2 cells, and other human mast cell lines available, we believe that LADR cells appear to resemble primary human cell cultures, with higher β-hex release upon FcεRI crosslinking, and a higher cellular content of tryptase perhaps due to their slower proliferation rates. HMC 1.1 and 1.2 cells lack functional FcεRI and ROSA^KIT WT^ cells lack chymase expression. Chymase is an efficient Ang II (angiotensin II)-forming enzyme and has been implicated in a wide variety of human diseases that also implicate its many other protease actions. LADR and LAD2 cells do not harbor mutations in *KIT*, expressed by as much as 80% of patients with systemic mastocytosis (SM), representing a disadvantage of this cell line for studying emerging therapeutics directed at *KIT* D816V. It should be recognized, however, that KIT-independent pathways and pro-oncogenic hits and lesions may be responsible for disease progression in advanced SM [5]. Thus, for example, due to upregulation of the viral oncogene c-MYC and expression of surface antigens CD13, CD33, CD34, CD117, CD123, and other markers aberrantly expressed on malignant mast cells and/or their progenitors, LADR cells may indeed be a tool for studying cell growth and regulation in advanced SM. The ability of human mast cells to act as a latent reservoir for HIV, as shown in vivo and with LADR cells, may be the best evidence that the LADR human mast cell line has wide-ranging utility and advantages well beyond the study of SM.

## 4. Materials and Methods

### 4.1. LAD Cell Cultures

LAD cells were maintained in StemPro serum free media with nutrient supplement, glutamine, penicillin/streptomycin, and 100 ng/mL rhSCF as described [7]. Hemi-depletions were performed weekly. Cell proliferation was examined by measuring total cell numbers weekly following staining with Kimura stain.

### 4.2. β-Hex Release Studies

β-hex release was assayed following crosslinking of FcεRI as described [8]. Briefly, LADR and LAD2 cells were incubated overnight with 100 ng/mL human myeloma IgE (BD PharMingen, San Diego, CA, USA), biotinylated by the NIAID core facility, followed by crosslinking with 125 ng/mL streptavidin (SA) alone or in combination with rhSCF (Sigma-Aldrich, St. Louis, MO, USA) [9]. β-hex was reported as a percentage of total cell contents released.

### 4.3. Flow Cytometry Studies

LADR and 2 cells were harvested, washed, and incubated with Aqua for to assess viability. Following additional washing, cells were incubated with either CD13, CD25, CD33, CD34, CD63, CD117, CD123, CD133, CD184, CD193, CD195, or FcεRI and examined. Alternatively, cells were fixed with 4% paraformaldehyde at room temperature for 10 min. After permeabilization with cold methanol for 1 h on ice, cells were stained with anti-tryptase for 30 min. For intracellular chymase, cells were fixed and permeabilized with Foxp3 staining buffer set (eBioscience, San Diego, CA, USA) and stained with anti-chymase for 30 min. All analyses were performed on at least 10,000 cells. All cells were analyzed by LSRFortessa flow cytometry.

### 4.4. Gene Arrays

LADR and 2 cells were harvested, and total RNA was extracted from 1 × 10^6^ LAD2 cells using the RNeasy Plus RNA isolation kit (Qiagen, Germantown, MD, USA) as described [10]. Approximately 1 µg of total RNA was reverse transcribed (RT) using the SuperScript III First-Strand synthesis system (Invitrogen, Grand Island, NY, USA) with random hexamer primers. cDNA and expression arrays comparing LADR with LAD2 gene expression were then prepared using RT-PCR according to manufacturers’ recommendations. Sequencing data was analyzed using Sequencher (Version 4.5, Softgenetics, State College, PA, USA).

### 4.5. HIV Infectivity

LADR and 2 cells were first incubated overnight with/without monomeric human IgE, washed, and examined for complement receptor expression over 96 h as previously described [6]. Stimulated cells were also examined for receptor mRNA expression over 96 h. Infectivity in cultured LADR and LAD2 cells with HIV-BAL, HIV-LAV, and HIV-TYBE was measured by assaying p24 expression [11].

## 5. Conclusions

LADR cells when compared to LAD2 cells have a slower proliferation rate, are larger, and more granulated. LADR cells have increased FcεRI/CD117 and tryptase expression and express chymase. LADR cells have different gene expression which is currently being investigated. Following monomeric human IgE treatment, LADR express CD184 and earlier infectivity with HIV-BAL, HIV-LAV, and HIV-TYBE. Thus, while LADR and LAD2 cells share many characteristics, in some situations, one cell line may have an advantage over the other.

## Figures and Tables

**Figure 1 ijms-20-05520-f001:**
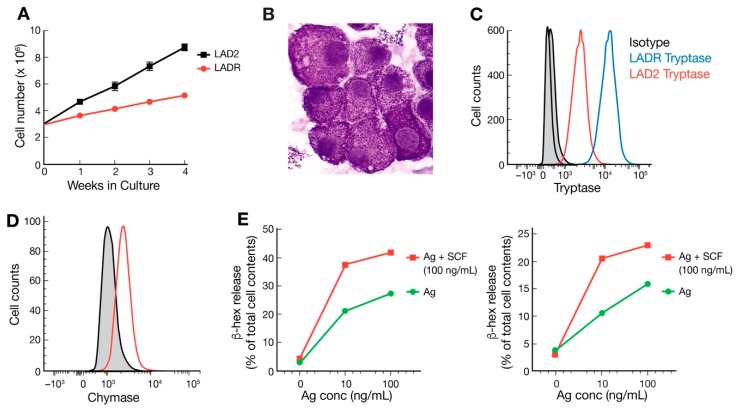
Cell proliferation, tryptase expression, chymase expression, degranulation, and beta-hexosaminidase (β-hex) release of LADR (a second mast cell line) and laboratory of allergic diseases 2 (LAD2) cells. (**A**) LADR cell numbers (in red) doubled in 3–4 weeks compared with 1–2 weeks for LAD2 cells (in black), LADR cells appeared to expand in culture as a more advanced human mast cell line; (**B**) Wright–Giemsa staining of LADR cells (×630); (**C**) LADR cells (in blue) have log-fold higher granular expression of tryptase; (**D**) LADR cells express chymase (in red, and (**E**) LADR cell β-hex release (left panel) was twice the release of LAD2 cells (right panel) following Ag crosslinking alone and with SCF (stem cell factor) enhancement.

**Figure 2 ijms-20-05520-f002:**
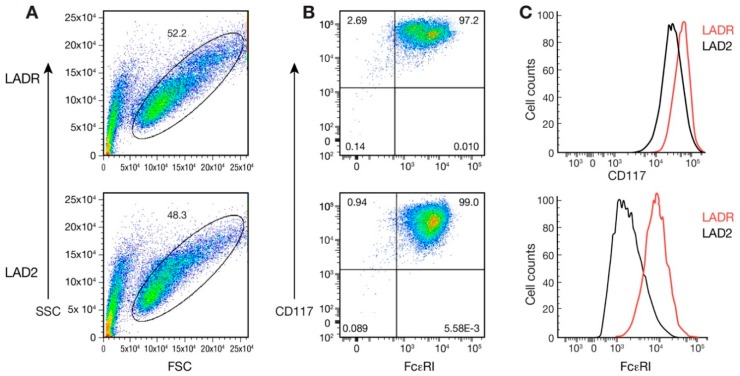
Flow cytometry studies comparing LADR with LAD2 cells. (**A**) LADR cells (upper panel) are larger (based on FSC, horizontal axis) and more granulated (based on SSC, vertical axis) when compared with LAD2 cells (lower panel). (**B**) LADR cells (upper panel) have higher expression of FcεRI (horizontal axis) and CD117 (vertical axis) when compared with LAD2 cells (lower panel), and (**C**) histograms of CD117 (upper panel) and FcεRI (lower panel) expression comparing LADR (in red) and LAD2 cells (in black) and consistent with results in B.

**Figure 3 ijms-20-05520-f003:**
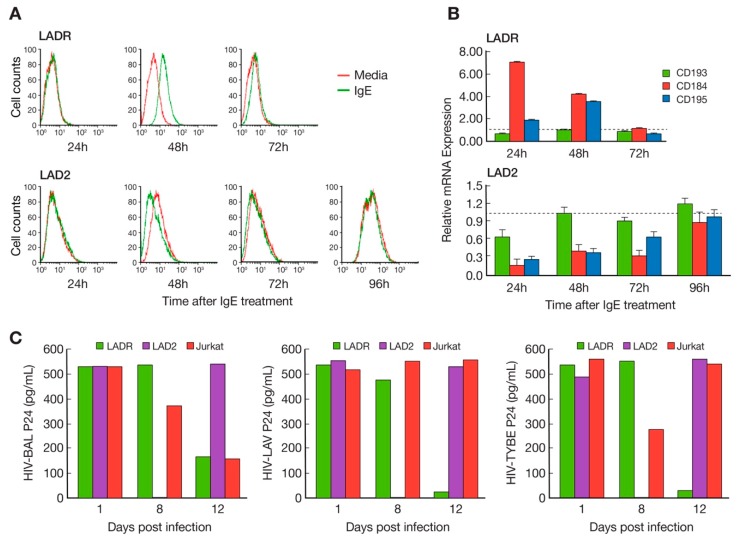
HIV infectivity studies comparing LADR and LAD2 cells. Following monomeric human IgE treatment, (**A**) LADR cells (upper panels) have higher surface expression of CD184 (green curve compared with red curve isotype control) than LAD2 cells (lower panels), consistent with (**B**) higher CD184 mRNA expression at 24 and 48 h, and (**C**) detectable infection (HIV-p24 assay) with HIV-BAL (left graph), HIV-LAV (middle graph), and HIV-TYBE (right graph) at day 8 for LADR cells (green) compared with day 12 for LAD2 cells (purple). The standard deviations in 3C for HIV-BAL, LAV, and TYBE p24 values all range from 0.05–0.1% and are not displayed.

**Figure 4 ijms-20-05520-f004:**
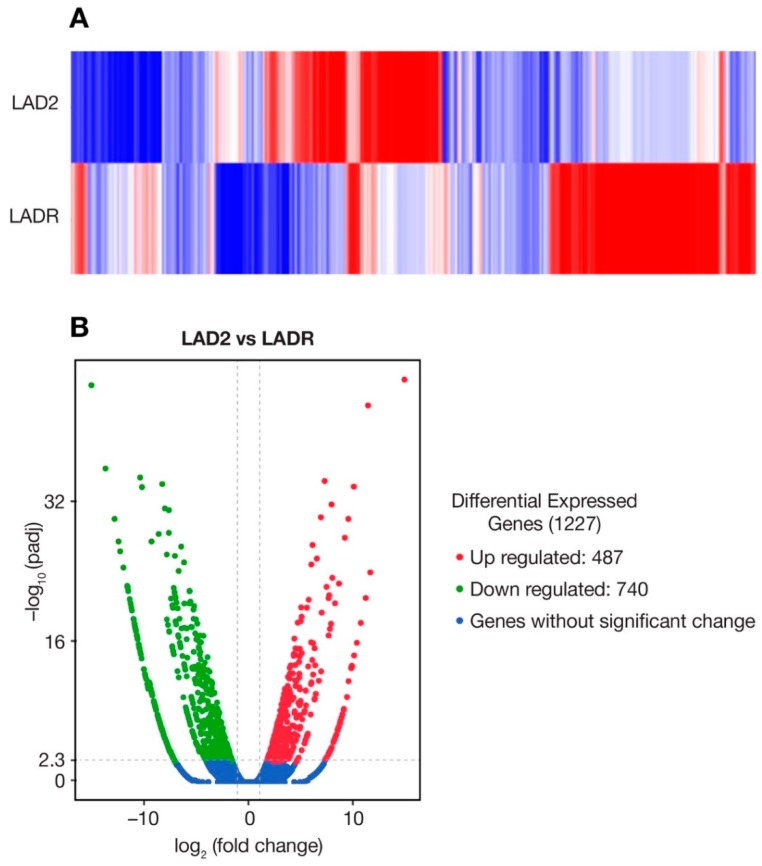
Differential gene expression of LADR cells compared with LAD2 cells. (**A**) Heat map showing differentially expressed and upregulated genes (red) when comparing LADR to LAD2 cells. Downregulated genes are shown in blue, genes without significant change are shown in white. (**B**) LADR cells expressed over 400 upregulated genes (red) when compared with LAD2 cells. Genes without significant change are shown in blue.

**Figure 5 ijms-20-05520-f005:**
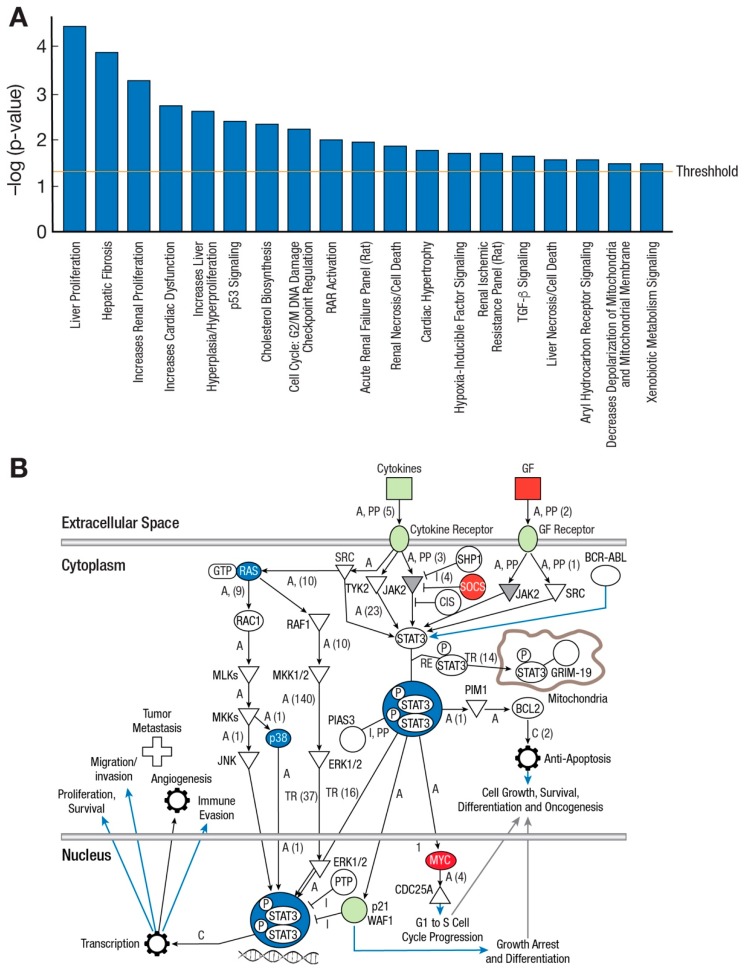
Canonical pathways and examples of gene upregulation in LADR cells. (**A**) Top canonical pathways for LADR cells compared with LAD2 cells, and (**B**) Suppressor of cytokine signaling (SOCS) genes are upregulated, which may contribute to an LADR cell inherent JAK2/STAT3 downregulation advantage. Upregulation of the viral oncogene *MYC* may provide an independent reason for cell proliferation, survival, and differentiation. Upregulated genes are highlighted in red. Black arrows designate pathways of interaction between signal transduction elements, blue arrows designate pathways of particular interest in the use of this cell line, T-arrow refers to inhibition.

**Table 1 ijms-20-05520-t001:** Surface expression of CD markers. LADR cells expressed CD13, CD33, CD34, CD63, CD117, CD123, CD133, CD184, CD193, and CD195, while LAD2 cells expressed CD33, CD34, CD63, CD117, CD133, and CD193 but not CD13, CD 25, CD123, CD184, or CD195.

CD Marker	LADR Cells	LAD2 Cells
CD13	+ +	−
CD25	−	−
CD33	+	+
CD34	+	+
CD63	+ +	+ +
CD117	+ + +	+ + +
CD123	+	−
CD133	+	+
CD184	+ +	−
CD193	+	+
CD195	+	−
FcεRI	+ +	+ +

Single multiplication sign (+) means 1 log increase over negative control, double sign (+ +) means 2 log increase over negative control, triple sign (+ + +) means 3 log increase over negative control. Negative expression is designated by (−).

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
