# Peer review of "Description and Characterization of a Novel Human Mast Cell Line for Scientific Study"

_ijms, 2019, doi:10.3390/ijms20225520_

Round 1

Reviewer 1 Report

The authors have written a clear and engaging paper on the development of a new cell line - a tool likely to be of good use to mast cell biologists.  The preliminary characterization they report is thorough and thoughtful.  In the version for peer review, Figures are hard to read - esp. Figures 4 and 5 and the resolution/quality could be improved.  Otherwise, great job! 

Author Response

Answers to Reviewer #1 Queries

The authors have written a clear and engaging paper on the development of a new cell line - a tool likely to be of good use to mast cell biologists.  The preliminary characterization they report is thorough and thoughtful.  In the version for peer review, Figures are hard to read - esp. Figures 4 and 5 and the resolution/quality could be improved.  Otherwise, great job!

Answer: My co-authors and I thank this reviewer for the positive comments. All figures have been reviewed and edited wherever needed for better resolution and quality. 

Reviewer 2 Report

In the submitted manuscript, Kirshenbaum and colleagues describe and characterize a novel human mast cell line, named LADR, as a new potential tool to improve the knowledge of human mast cell biology. The authors describe cell proliferation, expression of classical markers and some functional properties of the LADR in comparison with the previous described and worldwide adopted LAD2 cell line.

While the newly described cell line can be a new interesting tool for the MC scientific community, I found the functional characterization poor and not exhaustive (see below comment to Fig.1C). In addition, some data are indicated in the text but not provided and the quality of other figures is not sufficient for a proper evaluation.

Overall, from the data provided, I can’t really appreciate the advantages of this new cell line compared to previous described ones, as nicely reviewed in the cited Ref. 5, and its potential utility for the MC community. The authors suggest that LADR may be a “more advanced and mature cell line” and “appear to resemble primary human cell cultures”: a comprehensive functional parallelism between LADR and human MC primary cells (in addition to LAD2) would significantly increase the impact of this scientific contribution. Similarly, how “LADR cells may indeed be a tool for studying cell growth and regulation in advanced SM” should be discussed in more details in the conclusion section.

Comments:

The mutation status of Kit in LADR cells is not shown (only discussed);

L.59: “In culture, LADR cells were larger and more granulated”: a representative photomicrograph should be shown (also in consideration that flow data regarding the larger size and increased granularity of LADR (Fig. 2A) are not convincing and/or required further quantification);

L.62: Chymase staining is not provided;

Figure 1C: it would be worthy to show degranulation response to different stimuli other than IgE-dependent (i.e. ionomycin). Overall, other functional assays should be performed to improve the characterization of the cell line (i.e. cytokine release, chemotaxis, response to KIT inhibitors or other pharmacological targets for systemic mastocytosis);

Figure 2B: it looks like two populations with differential FceRI expression can be distinguished in the LADR cells histogram. Is that a relevant finding or an irrelevant observation? Is their functional response somehow affected/different?

Figure 3C: the resolution of the figure is insufficient. Honestly, I can’t read the legend and evaluate it. The same to be said of figure 5.

Figure 4A: it would be interesting to show the differential expressed genes in a table.

Table 1 legend is missing.

Author Response

 Answers to Reviewer #2 Queries

In the submitted manuscript, Kirshenbaum and colleagues describe and characterize a novel human mast cell line, named LADR, as a new potential tool to improve the knowledge of human mast cell biology. The authors describe cell proliferation, expression of classical markers and some functional properties of the LADR in comparison with the previous described and worldwide adopted LAD2 cell line.

While the newly described cell line can be a new interesting tool for the MC scientific community, I found the functional characterization poor and not exhaustive (see below comment to Fig.1C). In addition, some data are indicated in the text but not provided and the quality of other figures is not sufficient for a proper evaluation.

Answer: We do hope some will find this cell line a helpful tool in examining the biology of human mast cells. We have now provided additional data to support statements in the test and improved the quality of figures as requested

Overall, from the data provided, I can’t really appreciate the advantages of this new cell line compared to previous described ones, as nicely reviewed in the cited Ref. 5, and its potential utility for the MC community. The authors suggest that LADR may be a “more advanced and mature cell line” and “appear to resemble primary human cell cultures”: a comprehensive functional parallelism between LADR and human MC primary cells (in addition to LAD2) would significantly increase the impact of this scientific contribution. Similarly, how “LADR cells may indeed be a tool for studying cell growth and regulation in advanced SM” should be discussed in more details in the conclusion section.

Answer: The reviewer cites Ref. 5 as a good review of mast cell lines for the study of SM. The same Ref. 5 theorizes how KIT-independent pathways, pro-oncogenic hits and lesions and certain relevant CD markers may be responsible for disease progression in advanced SM [5].  We studied CD surface markers (Table 1) and pro-oncogenes (Fig. 5B) and found differences in LADR and LAD2 cells. Thus, as stated in our Discussion, p. 12, "Thus, for example, due to upregulation of the viral oncogene c-MYC and expression of surface antigens CD13, CD33, CD34, CD117, CD123, and other markers aberrantly expressed on malignant mast cells and/or their progenitors, LADR cells may indeed be a tool for studying cell growth and regulation in advanced SM." More research will be needed using LADR cells to determine if these surface antigens and oncogenic differences play a role in SM. 

Comments:

The mutation status of Kit in LADR cells is not shown (only discussed);

Answer: We now clearly state the cell line is derived from a patient with no mutations in KIT

L.59: “In culture, LADR cells were larger and more granulated”: a representative photomicrograph should be shown (also in consideration that flow data regarding the larger size and increased granularity of LADR (Fig. 2A) are not convincing and/or required further quantification);

Answer: As requested, a photomicrograph of LADR cells is now included in Fig. 1B.

L.62: Chymase staining is not provided;

Answer: As requested, intracytoplasmic staining for chymase is now included in Fig. 1D.

Figure 1C: it would be worthy to show degranulation response to different stimuli other than IgE-dependent (i.e. ionomycin). Overall, other functional assays should be performed to improve the characterization of the cell line (i.e. cytokine release, chemotaxis, response to KIT inhibitors or other pharmacological targets for systemic mastocytosis);

Answer: We appreciate the reviewer's comments. We successfully showed that LADR cells, as with LAD2 cells, have functional IgE receptors. More exhaustive experiments can be performed, but the data herein best characterizes this new functional cell line and its distinction from LAD2 cells.  Many of the questions imposed by this reviewer will be answered as we and others further examine this mast cell line. 

Figure 2B: it looks like two populations with differential FceRI expression can be distinguished in the LADR cells histogram. Is that a relevant finding or an irrelevant observation? Is their functional response somehow affected/different?

Answer: We have not yet studied subpopulations of LADR cells and the relationship of such observations to the maturation of these cells.

Figure 3C: the resolution of the figure is insufficient. Honestly, I can’t read the legend and evaluate it. The same to be said of figure 5.

Answer: As requested, we have improved the resolution and data presentation in Figs. 3C and 5.

Figure 4A: it would be interesting to show the differential expressed genes in a table.

Answer: Due to the large number of differentially expressed, upregulated genes in LADR cells, when compared with LAD2 cells, we elected to present canonical pathway differences (Fig. 5A) for the best overview and to prevent any bias for the reader. More research is needed to determine if upregulated genes in LADR cells have physiologic relevance in mast cell biology.

Table 1 legend is missing.

Answer: Legend is now added to Table 1 and reads:

"Table 1. Surface expression of CD markers. LADR cells expressed CD13, CD33, CD34, CD63, CD117, CD123, CD133, CD184, CD193 and CD195 while LAD2 cells expressed CD33, CD34, CD63, CD117, CD133, CD193 but not CD13, CD123, CD184 or CD195."

Round 2

Reviewer 2 Report

I would like to thank the authors for having taken into consideration my comments. The quality of the figures is definitively improved. However, I still have some concerns/comments about the data presented (see below). I maintain my original opinion that the quality and the relevance of the manuscript could be improved by performing some straightforward experiments (I refer here to non-IgE degranulation, chemotaxis, cytokine production) to better characterize and justify the use of LADR as a tool for studying human mast cell biology.

Figure 1 has been implemented with Wright-Giemsa and chymase staining of LADR cells. Since the figure is based on the comparison between LADR and LAD2, a representative photomicrograph (to justify the sentence “LADR cells were larger and more granulated”) and chymase staining for LAD2 cells should also be provided.

Table 1/Figure 3. In table 1, the authors suggest that LADR cells, but not LAD2, express CD184 and CD195 at surface level (in steady state?). However, in Figure 3A, CD184 is expressed by LADR only after 48h in the presence of IgE. Can the authors explain this apparent discrepancy? What about CD195?

Moreover, the authors state that “studies with LAD2 cells previously revealed that binding of monomeric human Ige to FcerI could affect HIV infectivity”. And in fact, in their previous publication (ref.6, Sunstrom et al., 2009), they wrote that LAD2 “expresses high levels of CD4, CCR5 and CXCR4”. I am wondering if it is a problem of different kinetics and/or experimental conditions, but the authors should clarify these contradictory findings about surface expression of CD markers in the two cell lines.

In fig. 1A, 1C, 3C, data are shown without any standard deviation. Similarly, statistical analyses are not provided. What are the data shown representative of? Numbers of replicates should be specified for each panel.

In the discussion, the author stated that: “LADR cells, similar to their LAD2 counterparts, survived for weeks in the absence of rhSCF, but did not proliferate” Is this data provided in the result section?

Considering the differential expressed genes (Fig 4 and 5), are the pathways underlined compatible with the more mature and differentiated profile of LADR cells? Can the authors comment on that?

Minor comments

I would suggest to adapt the same nomenclature for cell markers (i.e. CD184 or CXCR4) throughout the text and figures to simplify the reading. A description or reference for the p24 infectivity assay should be provided in the method section.

Author Response

I would like to thank the authors for having taken into consideration my comments. The quality of the figures is definitively improved. However, I still have some concerns/comments about the data presented (see below). I maintain my original opinion that the quality and the relevance of the manuscript could be improved by performing some straightforward experiments (I refer here to non-IgE degranulation, chemotaxis, cytokine production) to better characterize and justify the use of LADR as a tool for studying human mast cell biology.

Answer: As herein described, b-hex release with FceRI crosslinking is the characteristic of LADR cells which enables us to label it as an immunologically functional human mast cell line. In our experience with requests for LAD2 cells over 15 years, functional FceRI crosslinking has been the characteristic most sought after by researchers studying human mast cells. We anticipate once available that LADR cells will be studied in detail and that data from experiments describing non-IgE degranulation, chemotaxis and cytokine production will become available. We also note that in initial reports of other cell lines, such biologic responses as chemotaxis were not reported; and none examined HIV infectivity.

Figure 1 has been implemented with Wright-Giemsa and chymase staining of LADR cells. Since the figure is based on the comparison between LADR and LAD2, a representative photomicrograph (to justify the sentence “LADR cells were larger and more granulated”) and chymase staining for LAD2 cells should also be provided.

Answer: As initially requested, a photomicrograph of LADR cells and representative chymase flow staining have been added to Figure 1. A comparative photomicrograph of LAD2 cells was not added because our statement that LADR cells appeared larger and more granulated is based largely on a rightward shift in FSC and upward shift in SSC on flow cytometry. Judging size and granularity from two representative, histochemical staining images may allow for error or bias.

Table 1/Figure 3. In table 1, the authors suggest that LADR cells, but not LAD2, express CD184 and CD195 at surface level (in steady state?). However, in Figure 3A, CD184 is expressed by LADR only after 48h in the presence of IgE. Can the authors explain this apparent discrepancy? What about CD195?

Answer: Our response here has already been provided as an answer to the Academic Editor’s point/comment #1 and request for revision, and is repeated here:

“We suspect that previous flow cytometry experiments using our CD184 antibody were not sensitive enough to detect low level expression of CD184 on LADR cells at 24 hours in the presence of IgE. Experiments examining mRNA expression were performed at that same time and clearly show an increase in mRNA expression at both 24h and 48h. CD surface marker expression for LADR and LAD2 shown in Table 1 are very recent (2018), using newer conjugates and LSRFortessa flow cytometry. Thus, Discussion, lines 142-145, yellow highlight, have been modified and now read: "Previous flow cytometry experiments using CD184 antibody may not have been sensitive enough to detect low level expression of CD184 on LADR at 24 hours in the presence of IgE, in contrast to detection of CD184 expression using newer conjugates and LSRFortessa flow cytometry. Histograms depicting CD195 expression are not shown in Figure 3.”

Moreover, the authors state that “studies with LAD2 cells previously revealed that binding of monomeric human Ige to FcerI could affect HIV infectivity”. And in fact, in their previous publication (ref.6, Sunstrom et al., 2009), they wrote that LAD2 “expresses high levels of CD4, CCR5 and CXCR4”. I am wondering if it is a problem of different kinetics and/or experimental conditions, but the authors should clarify these contradictory findings about surface expression of CD markers in the two cell lines.

Answer: Similarly, our response here has already been provided as an answer to the Academic Editor’s point/comment #2 and request for revision, and is repeated here:

“Our current and previous publication using LAD2 cells refers to the experimental condition of receptor upregulation after incubation with IgE. This statement is made in the Results section, yellow highlight, lines 89-90. In the Discussion section, lines 136-137, green highlight, we again refer to our previous work that incubation of LAD cells with monomeric IgE may affect expression of CD4 and co-receptors. Discussion, lines 137-142, green highlight, describe our findings in Figure 3 and are as follows: "Functional differences were predicted when earlier and higher expression of CD184 and CD193 was noted for LADR cells following monomeric human IgE incubation (Fig. 3). Thus, when LADR cells were infected with HIV with different tropism, HIV-BAL, HIV-LAV and HIV-TYBE infectivity, as measured by p24 assay, were detected earlier than LAD2 cells, consistent with their differential surface expression of complement receptors.”

In fig. 1A, 1C, 3C, data are shown without any standard deviation. Similarly, statistical analyses are not provided. What are the data shown representative of? Numbers of replicates should be specified for each panel.

Answer: Similarly, our response here has already been provided as an answer to the Academic Editor’s point/comment #3 and request for revision, and is repeated here:

“Standard deviations are shown in Fig. 1A but are very small. Figure 3C standard deviations are all less than 1% of the mean (range from 0.05-0.1%) and will not display on this graph. Legend for 3C, lines 104-105, yellow highlight, has been modified and now reads: "...Standard deviations in 3C for HIV-BAL, LAV and TYBE p24 values all range from 0.05-0.1% and are not displayed."

In the discussion, the author stated that: “LADR cells, similar to their LAD2 counterparts, survived for weeks in the absence of rhSCF, but did not proliferate” Is this data provided in the result section?

Answer: We did not provide this data because, as expected, LADR cells share this characteristic with LAD2 cells and we published the original observation with LAD cells back in 2003. A statement confirming that LADR cells survive, but do not proliferate, in the absence of rhSCF should be sufficient for reviewers.

Considering the differential expressed genes (Fig 4 and 5), are the pathways underlined compatible with the more mature and differentiated profile of LADR cells? Can the authors comment on that?

Answer: LADR cells are too new of a cell line for us to comment or speculate, but we suspect that physiologic relevance of statistically significant upregulated genes and associated canonical pathways, when compared with LAD2 cells, will become more evident with additional research.     

Minor comments

I would suggest to adapt the same nomenclature for cell markers (i.e. CD184 or CXCR4) throughout the text and figures to simplify the reading. A description or reference for the p24 infectivity assay should be provided in the method section.

Answer: As requested, the Nomenclature has now been unified throughout the paper and figures. A new reference #11, yellow highlight, has been added for the p24 infection assay as follows:

Sundtrom J.B.; Ellis J.E.; Hair G.A.; Kirshenbaum A.S.; Metcalfe D.D.; Yi H.; Cardona A.C.; Lindsay M.K.; Ansari A.A. Human tissue mast cells are an inducible reservoir of persistent HIV infection. Blood 2007, 109, 5293-5300.